


Published on behalf of


# *Staphylococcus aureus* and methicillin-resistant *Staphylococcus aureus* in juvenile green turtle (*Chelonia mydas*) carcasses, rearing seawater, feed and their antibiotic resistances

Thanaporn Chuen-im[1], Korapan Sawetsuwannakun[1], Thongchai Taechowisan[1] and Nakarin Kitkumthorn[2]

[1] Department of Microbiology, Faculty of Science, Silpakorn University, Nakhon Pathom, Thailand
[2] Department of Oral Biology, Faculty of Dentistry, Mahidol University, Bangkok, Thailand

Corresponding author
Thanaporn Chuen-im,
chuenim_t@su.ac.th

## ABSTRACT

*Staphylococcus aureus* is an opportunistic bacterium that can infect humans and animals. We previously reported that *Staphylococcus aureus* as one of the most frequent Gram-positive bacteria found in the infection in juvenile green turtles (*Chelonia mydas*) from the Sea Turtle Conservation Center of Thailand (STCCT), Sattahip, Chonburi Province. It was also the most detected Gram-positive bacteria in rearing seawater. In this study, we investigated the presence of *S. aureus* in coastal seawater used as supply water to rearing containers, rearing water, fish fillet used as feed, and juvenile green turtle carcasses at STCCT. From the results, *S. aureus* can be isolated from rearing water, fish fillet, and juvenile turtle carcasses but not from incoming coastal seawater. The determination of antibiotic resistance against 11 drugs demonstrated that more *S. aureus* from juvenile turtles were antibiotic resistant than the isolates from rearing water and fish fillet. Furthermore, a higher isolate number of methicillin-resistant *S. aureus* (MRSA) was found in juvenile turtle carcasses. We also detected penicillin-susceptible MRSA and *mecA*-positive methicillin-susceptible *S. aureus* from juvenile turtles and fish fillet, respectively. Differences in the antibiotic resistance profiles were observed in this study compared with our previous observation. A change in the antibiotic resistance properties possibly continued in *S. aureus*. This finding suggests that the status of animal health is at high risk and emphasizes the need for a surveillance plan and treatment strategies to confront this serious threat.

## INTRODUCTION

*Staphylococcus aureus*, a Gram-positive bacterium, is extremely halotolerant (10% NaCl) with a wide range of growth temperature between 7 °C and 48 °C (*Schleifer & Bell, 2009*). Biochemically, this bacterium can ferment the sugar mannitol, which results in a pH change to acidic, and produces an important virulence factor, coagulase, in infection. *S. aureus* and *Staphylococcus* spp. were reported as the largest Gram-positive group in the bacterial flora of nesting green turtles at Tortuguero National Park, Costa Rica (*Santoro et al., 2006*). *Staphylococcus* species were also found to be one of the bacterial taxas isolated from the gastrointestinal tract of Hawaii green turtles (*Kittle et al., 2018*). This outcome suggests that *S. aureus* is a common strain in the bacterial flora of green turtles. However, this bacterium is an opportunistic pathogen for humans and animals (*Haag, Fitzgerald & Penadés, 2019*). Previously, we examined bacterial infection in lesions of juvenile green turtles and hawksbill turtles at the Sea Turtle Conservation Center of Thailand (STCCT) and revealed that *Staphylococcus* spp. were the most frequent isolated strain among Gram-positive bacteria (*Chuen-Im et al., 2010*). Publications also reported the association of *Staphylococcus* spp. and *S. aureus* with lesions in stranded green turtles and loggerhead turtles (*Orós et al., 2005*). This result indicates that *Staphylococcus* species., particularly *S. aureus*, are major opportunistic pathogens that can cause serious diseases in sea turtles.

Antibiotic-resistant bacteria pose a serious threat to humans and animals. The increased health risk of *S. aureus* from being opportunistic bacteria to pathogens is considered when it exhibits antibiotic resistance. According to the *World Health Organization (2024)*, methicillin-resistant *Staphylococcus aureus* (MRSA) has been categorized under the high-priority group due to its potentially severe infections, with increasing trends in resistance and considerably difficult treatment. MRSA shows resistance at a minimum inhibitory concentration $\geq 4$ µ/mL oxacillin (*Siddiqui & Koirala, 2025*). Although, no sea turtle death caused by MRSA has been reported, several publications have indicated mortality in marine animals due to MRSA infection. MRSA has been claimed to be a cause of death in stranded and captive born bottlenose dolphins under human care (*Faires et al., 2009*; *Mazzariol et al., 2018*). MRSA was also isolated from captive walruses, stranded pilot whales, and wild harbor seals (*Faires et al., 2009*; *Hower et al., 2013*; *Rubio-Garcia et al., 2019*). The resistance of MRSA to antibiotics is associated with the *mecA* gene, a part of the staphylococcal chromosome cassette *mec*, which encodes the low-affinity penicillin-binding protein PBP2a (*Beck, Berger-Bachi & Kayser, 1986*). This gene was detected in 95% of isolates with methicillin resistance (*Wielders et al., 2002*). Subsequently, the detection of the *mecA* gene in *Staphylococcus aureus* can be used in the determination of methicillin resistance (*Beck, Berger-Bachi & Kayser, 1986*). Presently, studies on bacterial isolation and their antibiotic resistance properties in sea turtles were mostly based on Gram-negative bacteria (*Oliveira et al., 2017*; *Ebani, 2023*), and little information is available regarding Gram-positive bacteria. Therefore, this study aimed to investigate the presence of *S. aureus* in the seawater supply, rearing seawater, fish fillet, and juvenile green turtle carcasses at STCCT, and to determine their antibiotic resistance. The data will be fundamental information for the establishment of a management plan in the conservation program of this endangered species.

## MATERIALS AND METHODS

### Sample collection

The experiments were conducted during years 2022–2024. All samples were kindly provided by the Sea Turtle Conservation Center of Thailand (STCCT), Sattahip, Chonburi Province. The STCCT, operated under the Royal Thai Navy, has established an early intervention program for sea turtle conservation (*Chuen-Im et al., 2010*). According to the program, green turtle eggs will be collected from nests to incubate them in places safe from humans and animals. After hatching, juvenile turtles will be raised in captivity and released back to the sea at the age of about 4 months. Ten juvenile green turtles, raised at the STCCT were collected immediately after death and stored on ice during transportation to the laboratory. Gross necropsies were performed by examining the internal and external abnormal symptoms or the presence of lesions prior to microbiological examination. Two seawater sources were used in the study, *i.e.*, rearing water and water supply for juvenile turtle rearing. Rearing seawater samples were collected from cement tanks that were used to keep the collected juvenile turtle samples. The water supply for turtle rearing was incoming coastal seawater at a distance of about 10–20 m offshore. Water sample collection was carried out as previously described (*Chuen-Im et al., 2019*, *2021*). The samples were kept on ice during transportation to the laboratory. Fish fillet samples were collected five times separately and used in microbiological investigation. The fish meat was kept in a sterile plastic bag and placed on ice during transportation. This study was approved by the Silpakorn University Animal Ethics Committee (MHESI 8603.16/4291_2Nov2563 and MHESI 8603.16/0300_18Jan2567) and the Department of Fisheries, Ministry of Agriculture and Cooperatives (the Approval number 3/2566_17Nov2566).

### Isolation and identification of *Staphylococcus aureus*

Isolation of *Staphylococcus aureus* from the water samples was carried out by spreading 0.1 mL of each sample on baird parker agar (BPA). For juvenile turtle carcasses, internal organs, including the heart, kidney, liver, and contents of the stomach and the small intestine, were used for bacterial examination. Samples were prepared using the aseptic technique. After dissection, the organs were weighed and washed several times with sterile 0.85% (w/v) normal saline buffer before homogenization using a pestle and mortar. The ground tissue was resuspended in 0.85% normal saline buffer at a ratio of 1:10 (w/v) to prepare a $10^{-1}$ dilution. The $10^{-1}$ dilution was further diluted into $10^{-2}$ to $10^{-3}$ before being spread (0.1 mL) on BPA. Similar to the juvenile turtle sample preparation, approximately 10–15 g of fish fillet was homogenized using a pestle and mortar. The ground meat was mixed thoroughly before 1 g sample was resuspended in 0.9 mL 0.85% normal saline buffer to make a $10^{-1}$ dilution. Using the tenfold serial dilution technique, $10^{-2}$ and $10^{-3}$ dilutions were prepared prior to the spreading on BPA. All plates were incubated at 37 °C for 24–48 h. Gray-black colonies were isolated from BPA and then streaked onto nutrient agar (NA) for further pure culture. *S. aureus* was identified using the following biochemical tests: motility, growth on NaCl, decarboxylase test, hemolysis, oxidase, catalase, urease, oxidation/fermentation, carbohydrate fermentation (glucose, lactose, mannitol, and
sucrose), coagulase test, and Gram-staining and bacterial shape observation. All media were purchased from HiMedia Laboratories (Mumbai, India).

## Antibiotic susceptibility

The antibiotic susceptibility of S. aureus was determined through the disc diffusion method (*Clinical and Laboratory Standards Institute (CLSI), 2015*; *Chuen-Im et al., 2021*). Briefly, bacteria were cultured in nutrient broth at 37 °C overnight and then diluted to a concentration of $10^8$ CFU/mL. The bacterial suspension was swabbed on Mueller Hinton Agar (HiMedia Laboratories, Mumbai, India) and used for antibiotic resistance examination. Discs of 11 antibiotics were used in the study, including beta-lactam (penicillin (10 IU), ampicillin (10 μg), cefazolin (30 μg), and cefoxitin (30 μg); aminoglycosides: streptomycin (10 μg), kanamycin (30 μg), gentamicin (10 μg), amikacin (10 μg), and tobramycin (10 μg); and others: tetracycline (30 μg) and chloramphenicol (30 μg) (HiMedia Laboratories, Mumbai, India). *Staphylococcus aureus* (ATCC 25923) was used as the drug susceptibility control. All plates were incubated at 37 °C for 24 h. The inhibition zone was determined by measuring the diameter of the clear zone, including that of the disc, and recorded in millimeters. Multidrug-resistant bacteria were defined as bacteria resistant to at least one of the three drug classes.

The identification of MRSA and methicillin-susceptible S. aureus (MSSA) was performed in accordance with the *Clinical and Laboratory Standards Institute (CLSI) (2013*, *2021)*. Isolates that showed resistance to 30 μg cefoxitin disc (clear zone ≤21 mm) were reported as MRSA, and MSSA (clear zone >21 mm) was reported when an isolate showed susceptibility to cefoxitin.

## Amplification of *mecA* and *femA* genes *via* polymerase chain reaction (PCR)

S. aureus DNA was extracted using a modified protocol from the work of *Taechowisan, Mungchukeatsakul & Phutdhawong (2018)*. Briefly, a pellet of 1.5 mL overnight-cultured bacteria was resuspended in 600 μL Tris-ethylenediaminetetraacetic acid buffer, added with 3 μL 10% sodium dodecyl sulfate and 3 μL 20 mg/mL proteinase K, and incubated at 65 °C for 30 min. The mixture was then mixed with 600 μL phenol/chloroform/isoamyl alcohol (25:24:1) and centrifuged at 12,000 rpm for 5 min. The supernatant was transferred into a new tube, and an equal amount of absolute ethanol was added to precipitate DNA. The DNA pellet was obtained through centrifugation at 12,000 rpm for 5 min before rinsing with 1 mL 70% ethanol and air-drying. Finally, DNA was dissolved with TE buffer, and the concentration was adjusted to 100 ng/mL before storage at −20 °C until required.

To investigate the presence of *mecA*, which encodes for an alternative penicillin-binding protein PBP2a, and *femA*, which encodes for a protein precursor in peptidoglycan biosynthesis in S. aureus, in the DNA sample, we set up a mixture containing 1X polymerase chain reaction (PCR) buffer with $MgCl_2$, 0.5 μL 10 mM dNTP, 0.5 μL of 10 pg forward and reverse primers each, and 1 μL 100 ng/mL DNA template, adjusted with sterile $dH_2O$ to 25 μL. The PCR conditions were initially denatured at 95 °C, 5 min before being subjected to 35 cycles of denaturation at 94 °C for 30 s, annealing at 52 °C, 30 s,

**Table 1 Sequence of *mecA* and *femA* gene-specific primers used in this study (*Al-Talib et al., 2014*).**

| Gene | Sequence 5′ to 3′ | Product size |
| --- | --- | --- |
| *femA* forward | CGA TCC ATA TTT ACC ATA TCA | 450 bp |
| reverse | ATC ACG CTC TTC GTT TAG TT | |
| *mecA* forward | ACG AGT AGA TGC TCA ATA TAA | 293 bp |
| reverse | CTT AGT TCT TTA GCG ATT GC | |

extension at 72 °C for 1 min, and final extension at 72 °C for 5 min before holding at 4 °C. The PCR product was analyzed *via* ethidium bromide-stained 1% agarose gel electrophoresis. Table 1 shows the sequences of the gene-specific primers used in this study.

## Statistical analysis

To investigate the significant differences in the antibiotic resistance of *S. aureus* from the three sample sources, we analyzed the hypothesis test results using R stat for the Friedman test and PMCMRplus packages for the *post hoc* tests (*Pohlert, 2024*). Significant differences in antibiotic resistances of *S. aureus* from seawater, fish fillets, and juvenile turtle carcasses were reported when $p$-value < 0.01.

## RESULTS

### Detection of *Staphylococcus aureus* in seawater supply and juvenile green turtle rearing seawater and their antibiotic resistance

To investigate *S. aureus* in coastal seawater, which were used as input to the sea turtle containers, and rearing water, we used BPA to screen bacteria from water samples. A total of 240 colonies were randomly selected and streaked on NA to obtain a pure culture. Then, all isolates were identified using biochemical tests. From the total of 240 isolates, the highest identified bacterial number was that of *Staphylococcus* spp. (171 isolates; 71%) followed by those of *Staphylococcus aureus* (44 isolates; 18%) and *Micrococcus* spp. (21 isolates; 9%). The other two bacterial species found were *Bacillus* spp. and Gram-negative bacteria *Proteus* sp. (three isolates and one isolate, respectively). All 44 *S. aureus* isolates were from rearing water, and no *S. aureus* was isolated from coastal sea water (water supply). Results of the coagulase test indicated that all *S. aureus* isolates were coagulase positive.

The 44 *S. aureus* isolates were then examined for their antibiotic resistance to 11 drugs through the disc diffusion approach. From the results, the most frequent resistant antibiotic was penicillin in β-lactam class (23%) (Table 2), followed by chloramphenicol (14%), cefazolin (9%), and finally amikacin (2%). We found two isolates (SA211 and SA321) that were resistant to cefoxitin (Figs. 1B, 1C), which suggests that they may be MRSA compared with MSSA (isolate SAP210 in Fig. 1A).

In this study, 11 drugs belonging to four classes (β-lactam, aminoglycosides, protein synthesis inhibitor tetracycline and chloramphenicol) were used. Most isolates were sensitive to all tested antibiotics (25 isolates; 57%) (Fig. 2). Ten isolates (23%) were

**Table 2 Antibiotic resistance of *Staphylococcus aureus* from rearing seawater, fish fillet (feed), and juvenile green turtle carcasses at STCCT.** Percentage of isolates is in parentheses.

| Antibiotic phenotype* | PEN# | AMP# | CFZ# | FOX# | AMI# | KA# | S# | TOB# | GEN# | TE# | CMP# |
|---|---|---|---|---|---|---|---|---|---|---|---|
| *Rearing seawater* | | | | | | | | | | | |
| S | 34 (77) | 39 (89) | 38 (86) | 42 (95) | 43 (98) | 42 (95) | 41 (93) | 42 (95) | 42 (95) | 41 (93) | 38 (86) |
| I | 0 | 2 (4) | 2 (5) | 0 | 0 | 0 | 0 | 0 | 0 | 0 | 0 |
| R | 10 (23) | 3 (7) | 4 (9) | 2 (5) | 1 (2) | 2 (5) | 3 (7) | 2 (5) | 2 (5) | 3 (7) | 6 (14) |
| Total (isolates) | 44 | 44 | 44 | 44 | 44 | 44 | 44 | 44 | 44 | 44 | 44 |
| *Fish fillet* | | | | | | | | | | | |
| S | 9 (40) | 21 (91) | 23 (100) | 20 (87) | 23 (100) | 23 (100) | 23 (100) | 23 (100) | 23 (100) | 18 (78) | 14 (61) |
| I | 7 (30) | 0 | 0 | 0 | 0 | 0 | 0 | 0 | 0 | 0 | 0 |
| R | 7 (30) | 2 (9) | 0 | 3 (13) | 0 | 0 | 0 | 0 | 0 | 5 (22) | 9 (39) |
| Total (isolates) | 23 | 23 | 23 | 23 | 23 | 23 | 23 | 23 | 23 | 23 | 23 |
| *Juvenile green turtle carcasses* | | | | | | | | | | | |
| S | 11 (28) | 30 (75) | 30 (75) | 27 (68) | 31 (77) | 30 (75) | 29 (73) | 27 (87) | 37 (92) | 29 (73) | 31 (77) |
| I | 4 (10) | 3 (8) | 3 (8) | 0 | 2 (5) | 2 (5) | 0 | 0 | 1 (3) | 2 (5) | 1 (3) |
| R | 25 (62) | 7 (17) | 7 (17) | 13 (32) | 7 (18) | 8 (20) | 11 (27) | 4 (13) | 2 (5) | 9 (22) | 8 (20) |
| Total (isolates) | 40 | 40 | 40 | 40 | 40 | 40 | 40 | 31 | 40 | 40 | 40 |

**Note:**
*S, susceptible; I, intermediate; R, resistant. #PEN, penicillin; AMP, ampicillin; CFZ, cefazolin; FOX, cefoxitin; AMI, amikacin; KA, kanamycin; S, streptomycin; TOB, tobramycin; GEN, gentamycin; TE, tetracycline; CMP, chloramphenicol.

intermediate or resistant to one drug, whereas nine isolates were intermediate- and/or resistant to >1 drug. The highest antibiotic resistance number, 7 out of 11 drugs, was observed in isolates SAP215 and SA321. The multidrug-resistant SAP215 isolate was resistant to all aminoglycoside drugs tested (streptomycin, gentamycin, kanamycin, tobramycin, and amikacin) and two protein synthesis inhibitors; tetracycline and chloramphenicol. MRSA isolate SA321 also showed the highest antibiotic-resistant number (seven drugs). In addition to cefoxitin, this isolate was resistant to six other drugs, which belonged to the β-lactam (penicillin, ampicillin, and cefazolin) and aminoglycoside (gentamicin, kanamycin, and tobramycin) classes. The third-highest resistant antibiotic number (four drugs) was detected in another MRSA isolate, SA211, which was resistant to three drugs in the β-lactam class (penicillin, cefazolin, and cefoxitin) and tetracycline.

All 44 isolates that were biochemically identified as *S. aureus* were further investigated for the presence of peptidoglycan biosynthesis gene *femA*, and the penicillin-binding protein PBP2a gene, *mecA*, in their genome through PCR with specific-gene primers. Figure 1D shows the amplified DNA fragments obtained from two MRSA isolates SA211 and SA321. The results of 1% agarose gel electrophoresis showed PCR products with sizes of 450 and 293 bp obtained from both isolates, which indicates the presence of *femA* and *mecA*, respectively, in their genome but not in the case of *Staphylococcus* sp. (Fig. 1D; lane 5). From the 44 isolates, 43 and three gave PCR products from *femA*- and for *mecA*-specific primes, respectively.

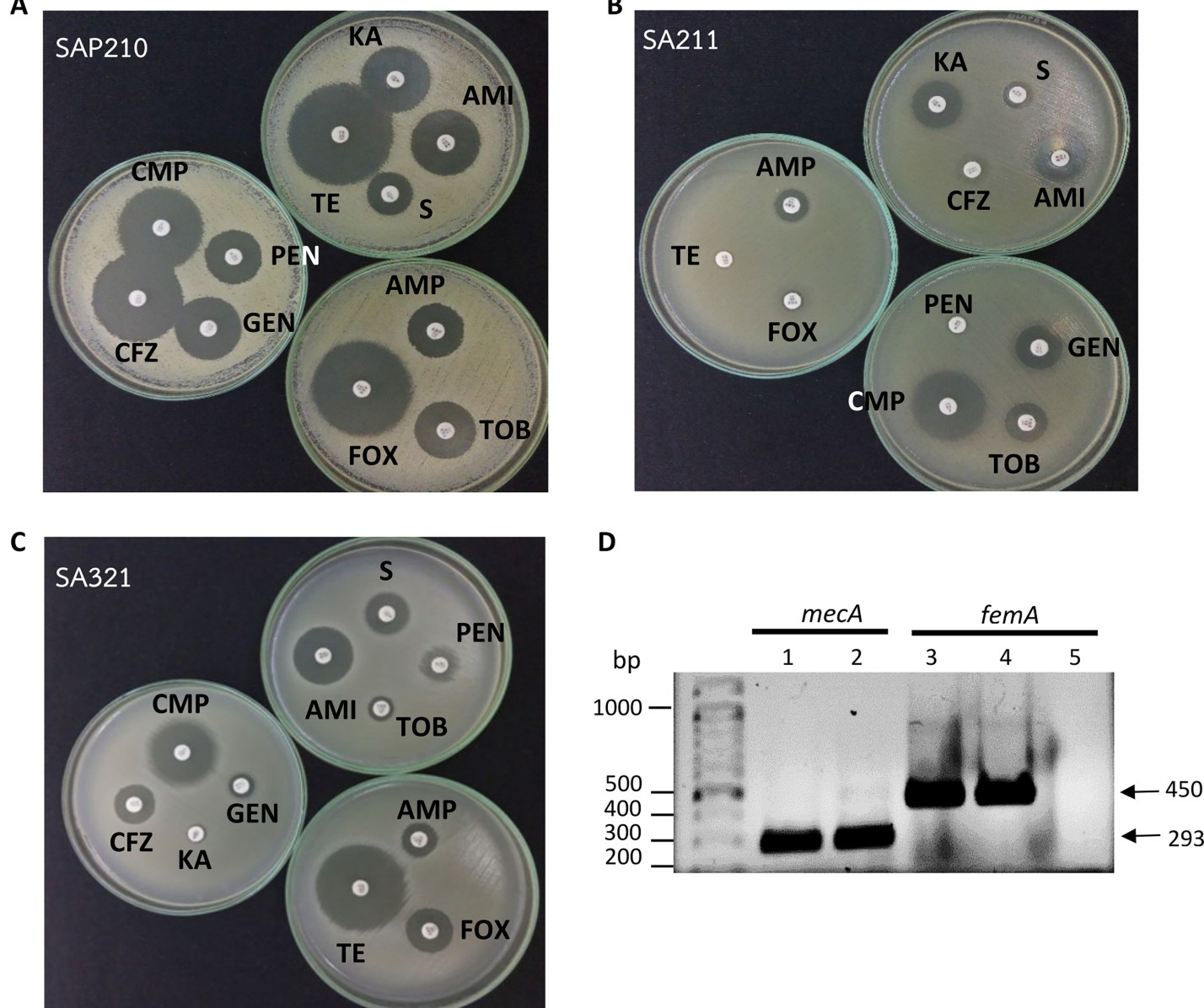

**Figure 1 Determination of methicillin-resistant *Staphylococcus aureus* (MRSA).** (A) Disc diffusion results obtained from methicillin-sensitive *S. aureus* (MSSA). (B) and (C) From MRSA. (D) PCR product obtained from *mecA* and *femA* specific primers; lanes 1 and 3: SA211, lanes 2 and 4: SA321, lane 5: *Staphylococcus* sp. PEN, penicillin; AMP, ampicillin; CFZ, cefazolin; FOX, cefoxitin; AMI, amikacin; KA, kanamycin; S, streptomycin; TOB, tobramycin; GEN, gentamycin; TE, tetracycline; CMP, chloramphenicol.

## Observation of *S. aureus* and MRSA in the feed of juvenile green turtles and their antibiotic resistance

As no *S. aureus* isolate was found in the sea water input to the rearing containers, we questioned whether the source of the bacteria detected in rearing water was the fish fillet used as feed. To investigate this issue, we used the fish fillet kindly provided by STCCT in

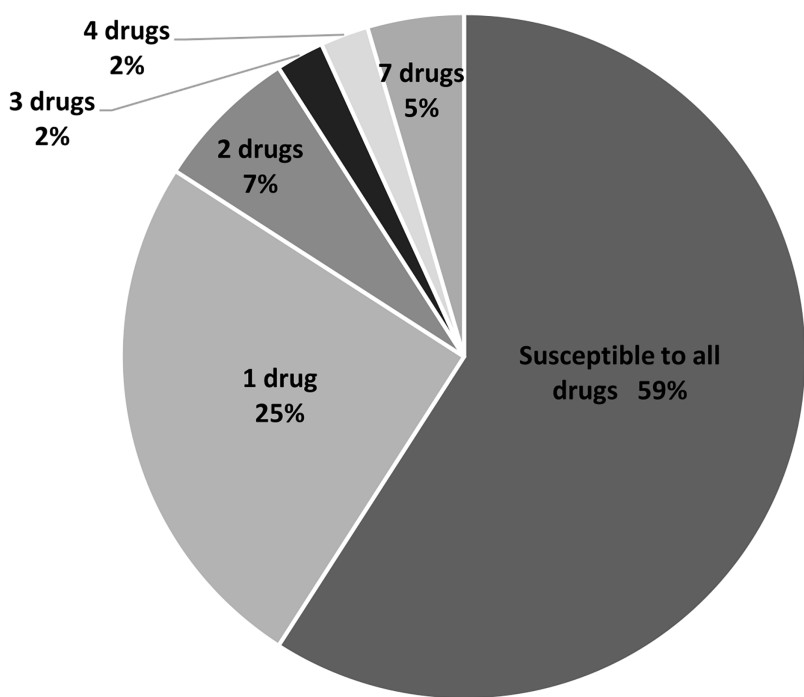

**Figure 2 Percentage and intermediate- or resistant-drug number of *S. aureus* isolates from rearing water (*n* = 44).**

isolation and identification. A total of 23 isolates were obtained from fish fillet in five samplings. All isolates were then subjected to disc diffusion and PCR experiments to determine the presence of MRSA. Table 3 shows the isolate numbers of cefoxitin-resistant or -susceptible bacteria and/or the detection of the *mecA* gene. From the 23 *S. aureus* isolates, 3 MRSA isolates were cefoxitin resistant and *mecA* gene positive, and 3 MSSA isolates were cefoxitin sensitive and *mecA* negative. However, 17 isolates that were positive in *mecA*-amplified PCR showed susceptibility to cefoxitin, which implies the presence of the *mecA* gene in their genome.

We further investigated the antibiotic resistance of these isolates to 10 other drugs. As displayed in Table 2, the *S. aureus* isolated from fish fillet showed resistance to five out of 11 drugs, which can be ranked in descending order as follows: chloramphenicol (39%), penicillin (30%), tetracycline (22%), cefoxitin (13%), and ampicillin (9%). From 23 isolates, five isolates (21%) were susceptible to all antibiotics, whereas the remaining isolates were mainly resistant to one drug and two drugs (eight isolates, 35% each). Finally, two isolates (9%) were resistant to three antibiotics. For MRSA, three isolates were resistant to cefoxitin (13%), and all isolates were resistant to penicillin. No multidrug resistance was observed.

## Investigation of *S. aureus* and MRSA in juvenile green turtle carcasses and their antibiotic resistance

To assess the health risk of juvenile green turtles from *S. aureus* infection, we investigated the presence of *S. aureus* and MRSA in juvenile turtle carcasses. Postmortem examination

**Table 3 Isolate number of *S. aureus* that showed cefoxitin resistant (R) or susceptible (S) and *mecA* negative (−) or positive (+).**

| Cefoxitin | R | R | S | S | Total |
|---|---|---|---|---|---|
| *mecA* | − | + | − | + | |
| Number isolates | 3 (13%) | 3 (13%) | 0 | 17 (74%) | 23 |

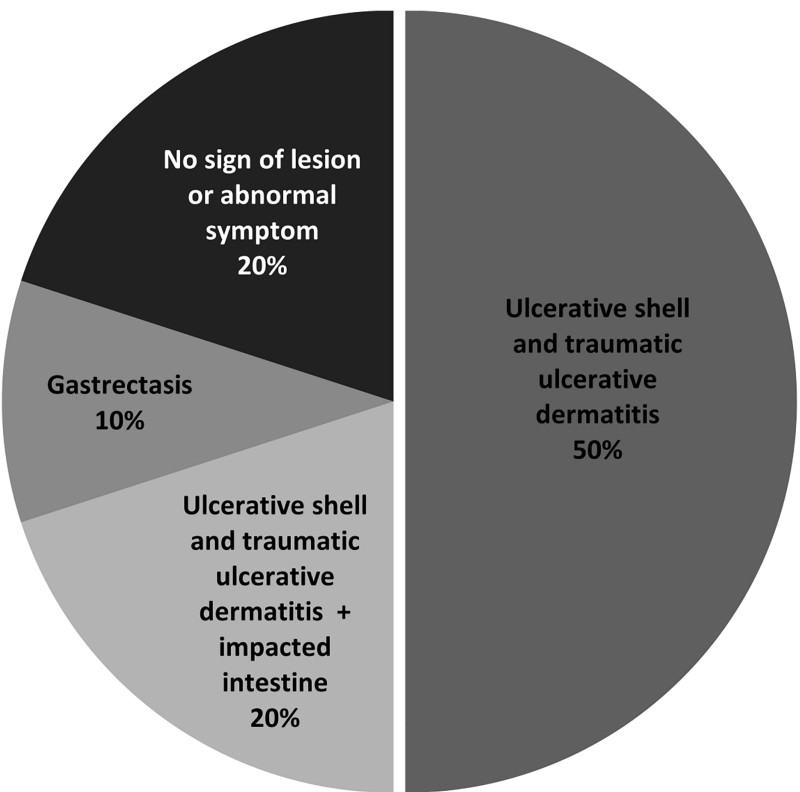

**Figure 3 Percentage of lesions or symptoms observed in ten juvenile green turtle carcasses.**

revealed that the primary lesion was ulcerative shell and traumatic ulcerative dermatitis (50%), mostly at the skins and appendages, whereas two out of 10 turtles (samples no. 5 and 7) showed no sign of lesions or abnormal symptoms (Fig. 3, Table 4). Afterward, the soft tissue organs of 10 turtles, which were the heart, liver, kidney, stomach, and small intestine, were used in bacterial isolation and identification. Black colonies with an opaque halo on BPA were further isolated for pure culture before identification using a biochemical test. *S. aureus* was isolated from seven out of 10 samples, which results in a total of 40 isolates (Table 4).

All turtle-isolated *S. aureus* were further examined for antibiotic susceptibility to 11 drugs. The results showed that five turtles (50%), which all exhibited lesions or abnormal symptoms, had MRSA (Table 4). When considering the percentage of isolates resistant to each antibiotic (Table 2), most were resistant to penicillin (62%), followed by those resistant to cefoxitin (32%) and then streptomycin (27%; 11 isolates).

**Table 4 Isolate number of S. aureus and MRSA detected in ten turtle samples.**

| Sample no. | Lesions/Symptoms | S. aureus | MRSA |
|---|---|---|---|
| 1 | Ulcerative shell and traumatic ulcerative dermatitis | 2 | ND |
| 2 | Ulcerative shell and traumatic ulcerative dermatitis + Impacted intestine | 7 | 2 |
| 3 | Ulcerative shell and traumatic ulcerative dermatitis | ND | ND |
| 4 | Ulcerative shell and traumatic ulcerative dermatitis | ND | ND |
| 5 | No lesion or abnormal symptom | ND | ND |
| 6 | Gastrectasis | 9 | 5 |
| 7 | No lesion or abnormal symptom | 10 | ND |
| 8 | Ulcerative shell and traumatic ulcerative dermatitis | 6 | 2 |
| 9 | Ulcerative shell and traumatic ulcerative dermatitis | 2 | 2 |
| 10 | Ulcerative shell and traumatic ulcerative dermatitis + Impacted intestine | 4 | 2 |
| Total | | 40 | 13 |

Note:
ND, not detected.

**Table 5 Antibiotic resistance profiles and multidrug resistance ($^M$) observed in S. aureus from rearing seawater, fish fillet, and turtle carcasses.**

| Pattern | Antibiotic resistance profile[#] | Isolate number | | |
|---|---|---|---|---|
| | | Seawater | Fish fillet | Turtle carcasses |
| 1 | PEN | 4 | 2 | 9 |
| 2 | CFZ | 1 | ND | ND |
| 3 | S | 1 | ND | 1 |
| 4 | TE | 1 | 2 | ND |
| 5 | CMP | 4 | 4 | 2 |
| 6 | PEN/AMP | 1 | ND | ND |
| 7 | PEN/FOX | ND | 1 | 1 |
| 8 | PEN/S | 1 | ND | ND |
| 9 | PEN/TE | ND | 2 | ND |
| 10 | PEN/CMP | 1 | ND | ND |
| 11 | CFZ/FOX | ND | ND | 1 |
| 12 | S/CMP | ND | ND | 1 |
| 13 | CMP/TE | ND | 5 | ND |
| 14 | PEN/CFZ/AMP | 1 | ND | ND |
| 15 | PEN/AMP/FOX | ND | 2 | ND |
| 16 | PEN/AMP/TE | ND | ND | 1 |
| 17 | PEN/AMP/CMP | ND | ND | 1 |
| 18 | PEN/S/TE$^M$ | ND | ND | 1 |
| 19 | PEN/CFZ/FOX/TE | 1 | ND | ND |
| 20 | PEN/CFZ/FOX/AMI/TE | ND | ND | 1 |
| 21 | PEN/FOX/S/KA/TE$^M$ | ND | ND | 1 |
| 22 | PEN/FOX/S/CMP/TE$^M$ | ND | ND | 1 |
| 23 | PEN/FOX/S/KA/AMI | ND | ND | 1 |
| 24 | PEN/CFZ/AMP/FOX/CMP | ND | ND | 2 |

| Pattern | Antibiotic resistance profile[#] | Isolate number | | |
| --- | --- | --- | --- | --- |
| | | Seawater | Fish fillet | Turtle carcasses |
| 25 | PEN/CFZ/AMP/FOX/S/KA/AMI | ND | ND | 1 |
| 26 | PEN/FOX/S/KA/TOB/AMI/TE[M] | ND | ND | 2 |
| 27 | PEN/CFZ/AMP/FOX/GEN/KA/TOB | 1 | ND | ND |
| 28 | S/GEN/KA/TOB/AMI/TE/CMP[M] | 1 | ND | ND |
| 29 | PEN/CFZ/AMP/FOX/S/GEN/KA/TOB/AMI | ND | ND | 1 |
| 30 | PEN/CFZ/AMP/FOX/S/GEN/KA/TOB/AMI/CMP/TE[M] | ND | ND | 1 |
| | Total (isolates) | 15 | 18 | 29 |

**Note:**
[M], multi-drug resistance; ND, not detected; [#]PEN, penicillin; AMP, ampicillin; CFZ, cefazolin; FOX, cefoxitin; AMI, amikacin; KA, kanamycin; S, streptomycin; TOB, tobramycin; GEN, gentamycin; TE, tetracycline; CMP, chloramphenicol.

Next, we compared the antibiotic resistance profiles of *S. aureus* from juvenile turtle carcasses with the isolates from rearing water and feed. Statistical analysis demonstrated a significant difference in the antibiotic resistance of *S. aureus* from the three sample sources ($p$-value = 0.003905). As provided in Table 5, for isolates with resistance to only one antibiotic, resistance to penicillin or chloramphenicol was observed in all sample sources. Most of the isolates from rearing water and fish fillet were resistant to one or two antibiotics, whereas almost half of the turtle isolates were resistant to >3 antibiotics, with one isolate showing resistance to all antibiotics tested. Moreover, six multidrug-resistant bacterial isolates were detected in juvenile turtle carcasses, and only one isolate was found in reared seawater. Of these, five isolates were MRSA (5 out of 7 isolates; 71%) and isolated from juvenile turtle carcasses. This finding indicates a severe situation of antibiotic resistance in *S. aureus* infection and a high risk of the health status of juvenile turtles at STCCT.

# DISCUSSION

This study investigated the presence of *S. aureus* in coastal seawater used as supplied water to juvenile turtle containers, rearing seawater, fish fillet as feed, and internal organs and gastrointestinal tract of juvenile green turtle carcasses at STCCT and their antibiotic resistance against 11 antibiotics. *S. aureus* was detected in rearing water, fish fillet, and turtle carcasses but not in incoming coastal seawater. This outcome is consistent with our previous report, which showed that *S. aureus* was detected in rearing water but not in coastal seawater near STCCT (*Chuen-Im et al., 2019*). This result suggests an additional origin of *S. aureus* in the rearing containers other than coastal seawater. Based on the results of this study, *S. aureus* can originate from fish fillet used as feed. Isolation of *S. aureus* in several marine animals, *e.g.*, dolphins, harbor seals, walrus, and grey seal, has also been reported (*Faires et al., 2009*; *Elk et al., 2012*; *Monecke et al., 2016*). This bacterium can either be introducing strains from human and terrestrial animals and environments or host-specific strains (*Elk et al., 2012*). However, the differences in the antibiotic resistance profiles observed in the isolates from turtle carcasses, rearing water, and fish fillet led to
another hypothesis that feed may not be the only possible sources. As STCCT is publicly accessible, that is, animal contact by visitors can occur, pathogenic agent transmission between human and animal may be possible (*Faires et al., 2009*; *Warwick, Arena & Steedman, 2013*).

We previously examined the antibiotic resistance of Gram-positive and -negative bacteria in juvenile turtle-rearing seawater and observed that the most and second-most frequently observed resistant antibiotics were ampicillin and penicillin, respectively (*Chuen-Im et al., 2019*). No isolate resistant to gentamicin was observed. In the present study, the most and second-most frequent resistant antibiotics were penicillin and chloramphenicol, respectively (Table 2). Two isolates were resistant to gentamicin together with another six antibiotics (Patterns 27 and 28, Table 5). Furthermore, the antibiotic resistance patterns of *S. aureus* in this study were different from our previous observation, which indicates changes in the antibiotic resistance of *S. aureus* and probably other bacteria species in rearing water. Most of the resistant antibiotics detected in turtle carcass isolates were not in direct contact with juvenile turtles at STCCT as proven by the presence of gentian violet, which is commonly used for the treatment of the infected turtles. This information led us to hypothesize that antibiotic resistance gene acquisition of bacterial flora in rearing water and juvenile green turtles possibly occurs through horizontal transferring *via* mobile genetic elements in the bacterial community (*Al-Bahry et al., 2009*). However, this hypothesis requires further investigation to draw a conclusion.

Presently, the observation of multidrug resistant bacteria in sea turtles has been increased in number and severity. Investigation of antibiotic resistance of bacterial isolates from loggerhead turtle in a rescue center in Southern Italy over the last decade revealed that four out of 138 isolates (2.9%) were resistant to all seven drugs tested (*Esposito et al., 2024*). The authors found an association of resistance with the frequency of drug used in treatment, which may subsequently lead to multidrug resistance. In this study, all three samples showed resistance to penicillin, a β-lactam drug. In 2006, *Harms et al. (2006)* studied the antibiotic resistance of bacterial flora from the cloaca of juvenile loggerhead turtles during 2004 and 2005 and observed that the highest antibiotic resistance was exhibited toward penicillin. *Al-Bahry et al. (2012)* investigated the antibiotic-resistant properties of bacterial isolates in the oviductal fluid of 20 female green turtles against four drugs, including ampicillin, streptomycin, sulfamethoxazole, and tetracycline. In their report, the β-lactam compound ampicillin was the most frequently detected drug in 42 resistant strains. Similarly, *Candan & Candan (2022)* conducted research to determine the resistance of bacteria isolated from nests and eggs, and observed that β-lactam was one of the main antibiotic resistance genes (ARGs) in these isolates. An investigation of the antibiotic resistance of bacterial isolates from the cloacal and nasal areas of green turtles admitted to a rehabilitation center revealed that the greatest resistance was observed with penicillin (*Tsai, Chang & Li, 2021*). For a decade, antibiotics in β-lactam class have been reported for high frequent resistance among bacterial isolates from sea turtles, suggesting that resistance to drugs in this class, particularly penicillin and ampicillin, can be a problem for sea turtle health.

Almost all MSRA isolates detected in this study were resistant to penicillin except one isolate from turtle carcass (Pattern 11 CFZ/FOX, Table 5). MRSA is normally resistant to β-lactam antibiotics including penicillin, due to the acquisition of the *mecA* gene, which encodes a low- affinity β-lactam-binding protein PBP2a (*Hartman & Tomasz, 1981*). Detection of this unusual penicillin-susceptible MRSA strain has been reported in patients who exhibited response to penicillin and clavulanic acid (Pen-Clav), a β-lactamase inhibitor, during their treatment for MRSA infection (*Harrison et al., 2019*; *Yan et al., 2024*). One explanation for this strain is that a mutation may occur in the *mecA* gene, which leads to a reduced gene expression and thereby reduced penicillin resistance (*Chen et al., 2022*). Alternatively, another mechanism of resistance, rather than a known mechanism, may be utilized in this strain (*Ryffel, Kayser & Berger-Bächi, 1992*). As the utilization of cefoxitin cannot distinguish penicillin- and clavulanic acid (Pen-Clav) resistant MRSA from Pen-Clav susceptible isolates, the double-disc diffusion method, 10 ug cefoxitin, and 1 ug oxacillin disc on Iso-Sensitest Agar must be used to differentiate between Pen-Clav resistant and -susceptible isolates (*Ba et al., 2022*). This rapid screening method may be useful for making decision in the treatment of animals infected with Pen-Clav susceptible isolates using potentiated penicillin.

Using the PCR approach, the investigation of the *mecA* gene in *S. aureus* genomic DNA revealed some PCR *mecA*-positive MSSA isolates (FOX-susceptible *S. aureus*) from fish fillet. The isolation of *mecA*-positive MSSA has also been reported in animals, including dogs and bovines (*Pu et al., 2014*; *Fabri et al., 2021*). To our knowledge, this study is the first report on *mecA*-positive MSSA isolation in green turtles. This *mecA*-positive methicillin-susceptible phenotypic strain was reported in a small percentage of MSSA isolated from patients (*Proulx et al., 2016*; *Liang et al., 2022*). Two mechanisms have been proposed to explain this strain: 1) inactivation of gene expression from the excision of transposable within the *mecA* gene, and 2) replication process errors due to the slip strand of genomic DNA (*Proulx et al., 2016*). The detection of *mecA*-positive MSSA should be considered a warning of a forthcoming resistant strain, because it can revert to MRSA during antibiotic treatment, particularly when using the combination of β-lactam compounds and a second drug (*Proulx et al., 2016*; *Amelia et al., 2024*).

As mentioned before, *S. aureus* is an opportunistic pathogen in humans and animals; however, the infection of antibiotic-resistant *S. aureus*, including MRSA may lead to an aggravated situation because it can result in increased treatment costs, a longer hospitalization period, and in worst cases, a high risk of treatment failure (*Inagaki et al., 2019*). Considering the number of antibiotics bacteria are resistant to, and multidrug resistance, *S. aureus* from juvenile green turtle carcasses exhibited more severe disease, if infected, than isolates from rearing water and fish fillet. Detection of MRSA in samples followed a descending order: juvenile green turtle carcasses (33%; 13 from 40 isolates), fish fillet (13%; three from 23 isolates), and rearing water (5%; 2 from 44 isolates). Further investigation in these isolates through the PCR method revealed several *mecA*-positive MSSA isolates in fish fillet and an unusual penicillin-susceptible MRSA strain in juvenile turtle carcasses. This finding demonstrates that alteration of antibiotic resistance continued in this bacterium. In this study, the importance of factors that should be

considered in breeding or rehabilitation centers for animals has been demonstrated. Nevertheless, other factors, if any, should not be excluded. Altogether, the infection of *S. aureus* can be a crucial health challenge for sea turtles, which emphasizes the need for a health management plan in terms of a monitoring plan and treatment strategies to fulfill the conservation program for these endangered species. The limitation of this study was the small sample size of juvenile turtles; higher sample numbers may be required for investigation to confirm the results. In addition, whether this bacterium is an active pathogen or an opportunistic infection, remains to be further investigated.

## CONCLUSIONS

*Staphylococcus aureus* is an opportunistic bacterium that can infect humans and animals. The risk increases for infection with MRSA because this strain can be resistant to multiple antibiotics. This study evaluated the risk of *S. aureus* on juvenile green turtles at STCCT. *S. aureus* can be isolated from rearing water, fish fillet, and juvenile turtle carcasses but not from incoming coastal seawater. The determination of antibiotic resistance demonstrated higher numbers of resistant antibiotics in isolates from juvenile turtles than in isolates from rearing water and fish fillet. A higher isolate number of MRSA was also found in juvenile turtle carcasses. Comparison of the antibiotic resistance profiles of the isolates between this study and our previous observation indicated that changes in the antibiotic resistance properties continued to occur in *S. aureus*. Our results suggest that the status of animal health is at high risk, which emphasizes the need for a surveillance plan and treatment strategies to confront this serious threat. Furthermore, sea turtles are possibly a reservoir of resistant bacteria in the marine environment. However, whether antibiotic-resistant gene transfer occurs in the bacterial flora of the sea turtle and the underlying mechanisms remain to be further investigated.

## ACKNOWLEDGEMENTS

We would like to thank the Sea Turtle Conservation Center operated by the Air and Coastal Defense Command, Royal Thai Navy, for their kindness in providing samples in this study. We also thank to Dr. Kannigar Hirunkasi for help on statistical analysis.

### Funding

This research was funded by the Faculty of Science, Silpakorn University, Grant no. SRIF-JRG-2567-09. There was no additional external funding received for this study. The funders had no role in study design, data collection and analysis, decision to publish, or preparation of the manuscript.

### Grant Disclosures

The following grant information was disclosed by the authors:
Faculty of Science, Silpakorn University: SRIF-JRG-2567-09.

## Competing Interests

The authors declare that they have no competing interests.

## Author Contributions

- Thanaporn Chuen-im conceived and designed the experiments, performed the experiments, analyzed the data, prepared figures and/or tables, authored or reviewed drafts of the article, and approved the final draft.
- Korapan Sawetsuwannakun performed the experiments, prepared figures and/or tables, and approved the final draft.
- Thongchai Taechowisan conceived and designed the experiments, analyzed the data, authored or reviewed drafts of the article, and approved the final draft.
- Nakarin Kitkumthorn conceived and designed the experiments, analyzed the data, authored or reviewed drafts of the article, and approved the final draft.

## Animal Ethics

The following information was supplied relating to ethical approvals (*i.e.*, approving body and any reference numbers):

Silpakorn University Institute Animal Ethics Committee (MHESI 8603.16/4291_2Nov2563; MHESI 8603.16/0300_18Jan2567).

## Field Study Permissions

The following information was supplied relating to field study approvals (*i.e.*, approving body and any reference numbers):

Fish fillet and juvenile green turtle carcasses were kindly provided by the Sea Turtle Conservation Center, the Air and Coastal Defense Command, Royal Thai Navy, and Certificate of Permission to conduct a research on animal species in the protected wildlife list under the Wildlife Preservation and Protection Act of 2019.

## Data Availability

The raw data for antibiotic resistance of each isolate and the DNA sequences of PCR product using *mecA*-specific primer are available in the Supplemental Files.

## Supplemental Information

Supplemental information for this article can be found online at http://dx.doi.org/10.7717/peerj.19579#supplemental-information.

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
