# Peer review of "Staphylococcus aureus and methicillin-resistant Staphylococcus aureus in juvenile green turtle (Chelonia mydas) carcasses, rearing seawater, feed and their antibiotic resistances"

_PeerJ, doi:10.7717/peerj.19579_

## Round 0.1 · original submission · Major Revisions

Dear authors,

Thank you for considering your work for PeerJ. Three experts have assessed your work, and the overall comments are positive. Two issues need your attention. First, please prove that the mecA amplicons are the target amplicon. Second, please check the manuscript for clarity and address all the comments raised by the reviewers. I also think that the manuscript is interesting and suitable for PeerJ.

Thank you so much for considering PeerJ for your research.

Good luck with your research moving forward.

Sincerely,
Bernardo

Reviewer 1 ·

Basic reporting

no comment

Experimental design

no comment

Validity of the findings

no comment

Additional comments

In the manuscript :Staphylococcus aureus and methicillin-resistant Staphylococcus aureus in juvenile green turtle (Chelonia mydas) carcasses, rearing seawater, feed and their antibiotic resistances, the authors describe an infection of green turtles at a conservation center in thailand. the turtles are apparently being infected with Staphylococcus aureus.

The authors report that possibly the origin of the infection is the turtles themselves and to a lesser extent Staphylococcus aureus is present in the tanks where the animals are kept and in the feed given to the turtles. Some questions arise from the results shown.

Where do the turtles from this center come from? do they reproduce there? did the turtles examined die as a consequence of Staphylococcus aureus infection?

Is there no prevention or quarantine mechanism to avoid the spread of the infection?

the infection only affects young (juvenile) turtles?

are these strains of Staphylococcus aureus common bacteria in these turtles?,is it possibly the captivity (under human care) that causes the infection?, are there immunogenic problems in the turtles that do not allow them to fight the infection?.

as it is a conservation center and a touristic place (pictures can be found on the internet) Is it possible that human contact is the source of infection?


the study shows the importance of considering different factors that have to be taken into account when having breeding centers for different animals.


in figure 1, specify clearly the abbreviations of the antibiotics used in the antibiograms, mecA and femA in italics.

347: MSRA is a MRSA type?

Reviewer 2 ·

Basic reporting

This article provides a valuable review of antimicrobial resistance in Staphylococcus aureus in sea turtles. It addresses a relevant and timely topic, especially considering the global concern about antimicrobial resistance and its links to human activities. However, there are some areas where the manuscript could be clearer and stronger to improve its overall quality, as outlined below:

Experimental design

Line 109
“Ten juvenile green turtles” "Why are there only 10 juvenile green turtles? The relatively small sample size raises questions about how broadly the findings can be applied. Explaining the reasoning behind the sample size. Is it because only these turtles could be collected during that period?

Line 128-129
Were these animals given antimicrobial biologics before they died? If so, the catology of the agents should be clarified.

Line 151-155
How antimicrobial agents were chosen/sellected and considered in the study would add clarity.

Validity of the findings

Line 306-307
“However, the differences of antibiotic resistance profiles observed in the isolates from turtle carcasses, rearing water, and fish fillet…”

The "differences" should be supported by statistics. For example, Fisher's exact test or chi-squared test can be used to test for significant or nonsignificant differences between sources (seawater rearing, pillet and carcasses) of all antibiotic resistances in Staphylococcus aureus.

Additional comments

Line 69
Type “(Chelonia mydas)” after “green turtles”.

Line 70-71
Replace “Staphylococcus species was also found to be one of bacterial taxon isolated from gastrointestinal tract of Hawaii green turtles” with “Staphylococcus species were also found to be one of the bacterial taxa isolated from the gastrointestinal tract of turtles.”

Line 72-73
“However, this bacterium is known as opportunistic pathogens for human and animals.” This sentence should be accompanied by appropriate references.

Line 78
Type “(Caretta caretta)” after “loggerhead turtles“.

Line 81
Replace “Antibiotic resistant bacteria are one of serious threats to human and animals. ”with “Antibiotic-resistant bacteria pose a serious threat to humans and animals”.

Line81-83
Replace “Increased health risk of S. aureus from opportunistic bacteria into pathogens is considered when it possesses antibiotic resistance.” With “The increased health risk of S. aureus from opportunistic bacteria to pathogens is considered when it has antibiotic resistance.”

Line 84
The definition of MRSA should be addressed here. For further details please see: https://www.ncbi.nlm.nih.gov/books/NBK482221/

Line 89
Please give more details for "human care", e.g. dolphins were captured in an aquarium or rescued from the wild.

Line 89-90 “MRSA was also isolated from walrus, pilot whales, and harbor seal”
Please specify the sources for walrus, pilot whales and harbour seals, e.g. wild or captive.

Line 128
"Why is only the baird parker agar (BPA) used? Are there any special requirements?"

Line 138-139
Replace “Gray-black colonies were isolated on nutrient agar (NA) for pure culture.” With “Gray-black colonies were isolated from BPA and then streaked onto nutrient agar (NA) for further pure culture”.

Line 156
“The inhibition zones were measured and recorded.”
How were the inhibition zones interpreted?

Line 162-163
“Isolate that showed resistant to cefoxitin disc will be reported as MRSA, and vice versa, MSSA will be reported when the isolate was susceptible to cefoxitin.”
The set should be accompanied by appropriate references and include the reference cut-off values used to classify each isolate as sensitive or resistant.

The English language should be improved to ensure that an international audience can clearly understand your text. Some examples where the language could be improved include lines 70-71, 81 and 81-83. I suggest you have a colleague who is proficient in English and familiar with the subject matter review your all manuscript, or contact a professional editing service.

Reviewer 3 ·

Basic reporting

The paper, titled "Staphylococcus aureus and methicillin-resistant Staphylococcus aureus in juvenile green turtle (Chelonia mydas) carcasses, rearing seawater, feed, and their antibiotic resistance," presents evidence of bacterial infection resistant to various antibiotics.

The work is well-written and easy to read, although its general conclusions leave inconclusive proposals. For example, they do not clarify whether these are infections caused by human handling or host-specific bacterial strains specific to aquatic animals because, as they mention, they are highly halotolerant bacterial species up to 10% salinity; therefore, they could be bacteria frequently present in the marine environment that spread in the presence of the Chelonid.

Experimental design

The authors determine the presence of the mecA gene by PCR as a determinant of methicillin resistance. However, they do not present evidence that at least some amplicons have been sequenced to prove that they come from this gene; they define it only by the expected molecular weight; confirmation by sequencing would be convenient.

Validity of the findings

In line 282, they describe the origin of S. aureus from juvenile turtle carcasses. However, they do not detail whether this bacterium is an active pathogen or an opportunistic infection. In previous work that they cite, Staphylococcus spp. had already been detected in unhatched Chelonia mydas eggs (Ebani, V. V. (2023)). Still, this bacterium is not described as pathogenic in living organisms.

The hypothesis that these are infections due to human handling, as a conclusion of the differences in antibiotic resistance profiles observed, is risky, and although it is under discussion because lateral transfers are carried out with high frequency in the marine environment (I recommend the reference https://doi.org/10.3390/d13090417)

---

## Round 0.2 · accepted · Accept

Dear authors,

Thank you so much for assessing all the concerns raised by the three experts who reviewed your manuscript. We all concur that the manuscript is now suitable for publication. Thank you so much for choosing PeerJ for your work.

All the best for your research moving forward.

Kind regards,
Bernardo

Reviewer 1 ·

Basic reporting

no comment

Experimental design

no comment

Validity of the findings

no comment

Additional comments

The authors of the study; Staphylococcus aureus and methicillin-resistant Staphylococcus aureus in juvenile green turtle (Chelonia mydas) carcasses, rearing seawater, feed and their antibiotic resistances (#115160), answered satisfactorily to the questions I asked, so I have no further recommendations.

Reviewer 2 ·

Basic reporting

The questions/suggestions I made in this manuscript have been addressed by authors.

Experimental design

The questions/suggestions I made in this manuscript have been addressed by authors.

Validity of the findings

The questions/suggestions I made in this manuscript have been addressed by authors.

Additional comments

The questions/suggestions I made in this manuscript have been addressed by authors.

Reviewer 3 ·

Basic reporting

no coments

Experimental design

no coments

Validity of the findings

no coments

Additional comments

Regarding my observations on the work, I consider the authors to have responded adequately, primarily regarding the identity of the amplicon, to which they add sequencing. The remaining observations are adequately addressed, with one minor observation requiring further investigation, but not critical to the article presented. It concerns the characteristics of the pathogenic or opportunistic strain in turtles.